# Ultrasound in the Diagnosis of Non-Expandable Lung: A Prospective Observational Study of M-Mode, B-Mode, and 2D-Shear Wave Elastography

**DOI:** 10.3390/diagnostics14020204

**Published:** 2024-01-18

**Authors:** Jesper Koefod Petersen, Katrine Fjaellegaard, Daniel B. Rasmussen, Gitte Alstrup, Asbjørn Høegholm, Jatinder Singh Sidhu, Pradeesh Sivapalan, Oke Gerke, Rahul Bhatnagar, Paul Frost Clementsen, Christian B. Laursen, Uffe Bodtger

**Affiliations:** 1Respiratory Research Unit, Department of Internal and Respiratory Medicine, Zealand University Hospital, 4000 Roskilde, Denmark; jekp@regionsjaelland.dk (J.K.P.); kafj@regionsjaelland.dk (K.F.); dbrs@regionsjaelland.dk (D.B.R.); gahn@regionsjaelland.dk (G.A.); ahoe@regionsjaelland.dk (A.H.); jssd@regionsjaelland.dk (J.S.S.); 2Institute of Regional Health Research, University of Southern Denmark, 5000 Odense, Denmark; 3Section of Respiratory Medicine, Department of Medicine, Herlev and Gentofte Hospital, University of Copenhagen, 2900 Hellerup, Denmark; pradeesh.sivapalan.02@regionh.dk; 4Department of Nuclear Medicine, Odense University Hospital, 5000 Odense, Denmark; oke.gerke@rsyd.dk; 5Department of Clinical Research, University of Southern Denmark, 5000 Odense, Denmark; 6Academic Respiratory Unit, University of Bristol, Bristol BS8 1TU, UK; rahul.bhatnagar@bristol.ac.uk; 7Copenhagen Academy for Medical Education and Simulation (CAMES), University of Copenhagen and the Capital Region of Denmark, 2100 Copenhagen, Denmark; paul.frost.clementsen@regionh.dk; 8Department of Respiratory Medicine, Odense University Hospital, 5000 Odense, Denmark; christian.b.laursen@rsyd.dk; 9Odense Respiratory Research Unit (ODIN), Department of Clinical Research, University of Southern Denmark, 5000 Odense, Denmark

**Keywords:** non-expandable lung, thoracic ultrasound, indwelling pleural catheter, diagnostic test

## Abstract

Background: Non-expandable lung (NEL) has severe implications for patient symptoms and impaired lung function, as well as crucial implications for the management of malignant pleural effusion (MPE). Indwelling pleural catheters have shown good symptom relief for patients with NEL; hence, identifying patients early in their disease is vital. With the inability of the lung to achieve pleural apposition following thoracentesis and the formation of a hydropneumothorax, traditionally, chest X-ray and clinical symptoms have been used to make the diagnosis following thoracentesis. It is our aim to investigate whether ultrasound measurement of lung movement during respiration can predict NEL before thoracentesis, thereby aiding clinicians in their planning for the optimal treatment of affected patients. Methods: A total of 49 patients were consecutively included in a single-centre trial performed at a pleural clinic. Patients underwent protocolled ultrasound assessment pre-thoracentesis with measurements of lung and diaphragm movement and shear wave elastography measurements of the pleura and pleural effusion at the planned site of thoracentesis. Results: M-mode measurements of lung movement provided the best diagnostic ROC-curve results, with an AUC of 0.81. Internal validity showed good results utilising the calibration belt test and Brier test. Conclusion: M-mode measurement of lung movement shows promise in diagnosing NEL before thoracentesis in patients with known or suspected MPE. A validation cohort is needed to confirm the results.

## 1. Introduction

Non-expandable lung (NEL) describes the inability of the lung to fully expand and achieve full pleural apposition. This typically occurs following pleural inflammation and subsequent visceral pleural fibrosis, but can also result from endobronchial obstruction [1,2,3]. NEL presents in recurrent pleural effusions with breathlessness, chest pain, or cough, which limits fluid drainage during thoracentesis. NEL also limits the efficacy of pleurodesis procedures due to the absence of pleural apposition [4,5].

NEL often presents as suspected iatrogenic pneumothorax after thoracentesis, in the form of hydropneumothorax on chest X-ray (CXR) [6]. Predicting the presence of NEL prior to thoracentesis remains challenging, and there is no standard approach [7], although techniques have been proposed to identify the condition both before and during thoracentesis. Previous studies have typically used a combination of radiological findings and intra-procedure clinical symptoms, but definitions have been inconsistent [7,8,9,10].

Intermittent pleural manometry during drainage has been suggested as the diagnostic method of choice, and is used in some centres [7]. Several studies [11,12] have proposed manometric cut-off values diagnostic of NEL; however, discordance between radiological and pleural manometry readings are a factor. While a high degree of pleural apposition on chest radiography predicts successful pleurodesis [5,7,13,14], fewer studies using pleural manometry have compared the same outcomes [7,15]. The recent study by Chopra and colleagues [16] showed that 28% of patients with elevated pleural elastance had complete lung expansion following thoracentesis, and 34% of patients with incomplete lung expansion detected by post-thoracentesis CXR had normal pleural elastance.

TUS assessment prior to thoracentesis is now considered the standard of care, and is included in several national and international recommendations [17,18]. Research groups have proposed TUS as a novel method of identifying NEL [19,20] before thoracentesis. Several ultrasound-based physiological measurements have been developed, including diaphragmatic and pulmonary movement [21,22,23,24]. Lung movement assessed by M-mode has been proposed as a possible method of identifying NEL before thoracentesis [19]. In one study, reduced lung pulse measured with M-mode measurement resulted in sensitivity and specificity values of 50% and 85% [20]. Ultrasound 2D shear wave elastography (SWE) allows for the characterization of tissue according to acoustic resilience, with pathological tissue differing in its mechanic properties compared to the surrounding normal tissue. In general, stiffer tissue results in faster tissue shear wave propagation [25], and this technique has previously been applied to help differentiate benign from malignant breast masses [26], subpleural solid masses [27], pleural effusions, and hilar lymph nodes [28,29]. However, it has no role in current guidelines for the work-up of suspected NEL [18,30].

In this study, we hypothesize that decreased movement of the visceral pleura in the context of NEL will be detectable by ultrasound measurement. We also hypothesize that SWE is superior to B-mode and M-mode in the diagnosis of NEL in patients with suspected MPE as it differentiates normal from fibrotic pleural lining.

## 2. Materials and Methods

A prospective observational study was conducted at a single secondary care hospital in Denmark (Zealand University Hospital, Naestved). Patients were eligible if they were over 18 years old, with a pleural effusion of over 2 cm depth needing thoracentesis in the context of suspected or known malignancy. Exclusion criteria were an estimated life expectancy of less than three months or a contraindication to thoracentesis. All patients provided informed consent. The study was approved by the Regional Ethics Committee (approval nr. SJ-891) and Region Zealand Data Committee (REG-001-2021). All TUS examination and thoracenteses were performed by two experienced and certified operators (JKP and KF).

Patient examination followed the following steps: pre-procedure baseline data collection and ultrasound measurements at site of planned thoracentesis, incorporating B-mode, M-mode, and SWE. Thoracentesis was performed and the hemithorax was assessed by ultrasound for complete pleural fluid drainage status, defined as less than 0.1 L remaining fluid assessed by visual estimation, and pleural apposition. All patients had a CXR after thoracentesis, except if a chest computer tomography (CT) was booked by the treating oncologist or physician the same day. The presence or absence of NEL was recorded following the four-month follow-up period from thoracentesis using the electronic patient records.

### 2.1. Pre-Procedure Baseline Data

Patient demographics and patient questionnaires were collected before examination. Medical records were examined for causes of pleural effusion.

### 2.2. Ultrasound Examinations

Ultrasound assessments were performed using LOGIQ S8 (GE Healthcare, Wauwatosa, WI, USA) with a C1-6-D curved (2–5 Mhz) transducer in the abdominal preset. Patients were scanned while seated upright, leaning forward, and resting upon a support stand. To avoid exertion-induced changes in respiration rate, patients were given least 5 min of rest before the examination was performed.

All measurements were obtained from a designated optimal site of thoracentesis, allowing for some minor transducer movement to optimize imaging (e.g., moving out of rib shadow). In order to identify and mark a safe site for thoracentesis, the diaphragm was located, effusion size and characteristics (e.g., septations, depth of effusion) were noted, and proximity to organs was considered. All effusions were basal.

Pleural effusion was evaluated by pleural thickening (>10 mm), swirling and septations, atelectasis/consolidation of the lung, pleural nodules, and lung sliding apically to the effusion (present or non-present). Four-second movie clips were recorded for measurements.

### 2.3. B-Mode

Maximum diaphragm movement was assessed from US recordings, and diaphragm amplitude was measured in centimetres. The area method, as described by Skaarup et al. [24], was utilized, with a lateral, mid-axillary view targeting optimal diaphragm visualization. A film clip was saved, and images of maximal inspiration and expiration were identified. A region from above the diaphragm was traced, and the area was calculated using the area-function of the ultrasound machine. The maximum areas of inspiration and expiration were calculated and subtracted, resulting in a measurement of diaphragm movement.

### 2.4. M-Mode

Diaphragm movement was measured at the area of largest amplitude of the diaphragm with the targeting line. Lung movement (visceral pleura) was assessed at the perpendicular point of thoracentesis, as illustrated in Figure 1. Amplitude results are provided in centimetres with two decimals.

### 2.5. 2D Shear Wave Elastography (SWE)

Measurements from three structures were made: five recordings of the parietal pleura, the pleural effusion superficially to the lung, and the visceral pleura/consolidation perpendicular to the transducer (Figure 1). Care was taken to apply minimal pressure from the transducer to the chest wall. Readings were obtained while breath was held. A region of interest (ROI) was selected upon the elastogram, providing adequate shear wave propagation, and was adjusted to size. Measurements were obtained in m/s [31].

### 2.6. Thoracentesis

Thoracentesis was performed according to local guidelines, with US-guided administration of local anaesthesia in the skin and intercostal structures [17]. A 7 French pigtail catheter was inserted and connected to a sealed system. The effusion was drained under gravity until fluid cessation (with a pause for each 1000 mL fluid removed) or until the patient experienced discomfort, e.g., tightness of chest, dyspnoea, or pain. Drainage was discontinued if symptoms were not alleviated by a pause. At premature flow cessation, the drain was flushed with 50 mL saline to check for blockage. CXR was performed within 2 h after thoracentesis.

### 2.7. Scoring of Non-Expandable Lung (NEL)—Reference Standard

As per the method described by Salamonsen et al. [20], NEL was defined based on a combination of clinical observations and post-thoracentesis radiology (Table A3). Two interventional pulmonologists (AS and JS) independently assessed the study data after four months of follow-up. Post-thoracentesis radiology reports and images obtained both directly following thoracentesis and in the follow-up period were available, and assessors recorded whether they observed air (or hydropneumothorax) in the pleural space surrounding the lower lobe, with examples shown in Figure 2. The assessors were blinded to the results from the index tests, e.g., SWE, M-mode, and B-mode measurements. The outcome was classified as: NEL (definite NEL, probable NEL), expandable lung (EL) (probably expandable, definitely expandable), or “Unable to score”. In the event of disagreeing diagnoses between assessors, a third interventional pulmonologist (UB) assessed the study information and a consensus was achieved. All assessments were made while blinded to the other assessor’s results. The assessors were provided with clinical observations during thoracentesis: development of chest tightness or severe coughing not alleviated by thoracentesis pause, air in chest tube, or undrainable pleural effusion despite unblocked chest tube.

### 2.8. Statistical Analyses

Normality distribution was tested using density histograms and the Shapiro–Wilk test of normality for continuous variables. Normally and non-normally distributed variables were expressed as means (±standard deviation, SD) and medians (interquartile range 25–75%, IQR), respectively.

As our ultrasound measurements had not been tested before in the context of diagnosing NEL, we allowed for a broad approach to the statistical analysis. Receiver-operating characteristic (ROC) curves were constructed to calculate the areas under the curve (AUC) for M-mode (lung and diaphragm movement), B-mode (diaphragm movement, area method: delta area), and SWE (parietal pleura, pleural effusion, and visceral pleura).

Several methods were explored to approximate the most clinically relevant variables. A cut-off at minimum 80% specificity was chosen as clinically relevant. For an estimation of an optimal cut-off point, the Youden index was chosen [32] to maximise the difference between the sensitivity and the false positive rate. For the best-performing model, a Brier score was calculated to evaluate the *goodness of predicted probability*. A calibration belt plot for internal validity [33,34] was used to estimate the accuracy of the measurements compared to the true outcome observed in the data.

As the cohort was too small to be stratified into both development and validation cohorts, a logistic regression analysis was performed to propose an optimized model for a future validation study.

The null hypothesis was rejected at the level of 5% (*p* ≤ 0.05). Statistical analysis was performed using STATA (StataCorp. 2021. *Stata Statistical Software: Release 17*. College Station, TX, USA).

## 3. Results

Between May 2021 and December 2021, 132 patients were screened, and 49 patients were included in the study. Of those excluded, 22 had too little fluid, 34 had benign aetiologies, 13 had short life expectancies, 12 declined, and 2 could not give consent. Of the included patients, 14/49 (29%) were categorized as NEL (Table 1). The mean age was 71 years (SD 9), and 25 patients were male (51%). The average volume of pleural fluid removed was 1178 mL (SD 629 mL), and the majority (*n* = 48; 98%) had post-thoracentesis radiological examinations immediately (Table 1). All ultrasound tests showed right-skewed distribution and tested significantly for non-parametric distribution.

At follow-up, most (*n* = 44; 90%) of the patients were diagnosed with malignancy (lung cancer: 23; breast cancer: 11; mesothelioma: 4), and the remaining had pleural infection, chronic pleuritis, or congestive heart failure. In total, 13 patients with NEL had malignancy at follow-up: lung cancer (6), mesothelioma (4), and other cancers (3).

Concerning the diagnostic property of TUS in NEL, Table 2 shows the AUCs from all eight ultrasonographic tests. We found SWE to be inferior compared to M-mode of lung movement, M-mode movement, and B-mode of diaphragm movement (AUC 0.81, 0.77, and 0.65, respectively). The AUCs of SWE were 0.57, 0.47, and 0.59 for parietal pleura, pleural effusion, and visceral pleura, respectively. ROC curves are shown in Figure 3.

Table A1 shows the ideal cut-point at which to optimize ultrasound diagnostic yield of TUS measures while allowing for at least 75% and 80% sensitivity and specificity. For each of the three best-performing tests, as given by the AUC, optimal cut-off points were calculated according to Youden’s index. The Brier score was calculated at 0.15, and the calibration belt plot showed a high degree of internal validity, with a test statistic of 0.08 and *p*-value of 0.78. As seen in Figure A1, the calibration belts encompassed the bisector at all times. This suggests that the predictions of the model did not significantly deviate from the observed findings in our developmental sample.

We found no diagnostic benefit of combining different TUS measures (Table A2). In univariate analyses, M-mode lung movement, M-mode diaphragm movement, and lung sliding were significantly associated with NEL. In multivariate regression, only M-mode lung movement remained significant (OR 7.72, 95% CI 0.84–70.90, *p* = 0.071). No combination of test modalities significantly increased the AUC above the ability of M-mode lung alone (Table A2).

## 4. Discussion

NEL represents a unique aspect in pleural disease pathophysiology, involving an increase in pleural elastance and pleural infiltration that results in a stiffer visceral membrane and restricted lung movement [1]. Cancer, which often results in malignant pleural effusion (MPE), is a leading cause of NEL, usually as a consequence of persistent pleural inflammation, disruption of the pleural barrier, atelectatic tumour obstruction, and pleural restriction [3]. Some studies have suggested that upwards of 30% [8,35] of patients with MPE are at risk of developing NEL, and that NEL may be the most frequent cause of pneumothorax following ultrasound-guided thoracentesis [11]. The high incidence of MPE and NEL [36] and the significant impact on quality of life of the affected patients warrant increased focus on these conditions.

Both MPE and NEL have been associated with a greater healthcare burden and shorter survival [37]; MPE’s median survival has been estimated at 5–8 months. Pleurodesis and placement of an indwelling pleural catheter (IPC) are the most common “definitive” procedures for patients with recurrent effusions in a cancer setting. For those patients with NEL, treatment of dyspnoea can be challenging, but in those who gain relief from thoracentesis, IPC has been proven to be the best treatment option [13,38,39]. Early detection of NEL facilitates optimal MPE treatment, which reduces symptom burden and time spent in hospital.

Ultrasonographic elastography was introduced in the 1990s and is now part of routine assessments of liver fibrosis and breast lesions [40]. However, it currently has no role in the work-up of suspected NEL [18,30]. In this first study on 2D shear wave elastography in the diagnosis of non-expandable lung, we found SWE to be inferior to both conventional M- and B-mode measurements of lung and diaphragm movement in predicting NEL. Lung movement measured by M-mode showed good predictive values on both ROC curves, with an AUC of 0.81 and clinical viable cut-off values as applied. Youden’s index showed promising results for sensitivity, specificity, PPV, and NPV of 0.78, 0.71, 0.52, and 0.89, respectively. Thus, our data supports the findings of both Salamonsen [20] and Flora [19], i.e., that hindered movement of the lung is a valid predictor of NEL. This also complies well with the pathophysiological description of NEL.

Table A4 lists selected cut points, allowing for an overview of the predictive capabilities of the measurements. Several methods exist for optimizing cut-off points, all with their individual trade-offs. Youden’s index is frequently used and presents a commonly used theoretical cut-off. Other methods, such as Liu’s and “nearest to (0, 1)”, showed the same cut-off and AUC (Table A3).

Our study, however, differs in several ways from those conducted previously [20]. In the Salamonsen study, TUS was performed by cardiologists or pulmonologists trained by cardiologists. Additionally, the ultrasound measurements which were relied upon (e.g., STI strain analysis) and subsequent calculations were not part of the standard training of the respiratory ultrasound practitioners who typically manage patients with NEL. We present data on a widely available and easily implementable method of measuring lung movement which has shown good discrimination between expansive lung and NEL, and which does not require any further specific training for either experienced professionals or US operators with basic 2D and M-mode competency in TUS.

There are no gold-standard, or even readily agreed-upon, diagnostic criteria for NEL. Clinical symptoms such as excessive coughing, increasing dyspnoea, and general discomfort due to reduction in intrathoracic pressure are all well recognized symptoms during thoracentesis, and may be suggestive of NEL. Radiological signs such as thickened visceral pleura and (hydro)-pneumothorax are most readily distinguishable on post-thoracentesis imaging. They are strongly suggestive of NEL and are usually what is relied upon in the clinical setting. These findings are well established, but are obtained after or during thoracentesis. This presents a potential risk of delaying or overlooking NEL. Pre-existing knowledge of NEL has the potential to alter both procedural approaches and treatment options for those with effusions. Several studies [10,41,42] have suggested pleural manometry as a feasible and viable standard for diagnosing NEL. The standardization of both methods and cut-offs has been suggested, although these vary according to the study, and as of yet, guidelines have not proposed pleural manometry as the reference standard nor shown the ability to prevent pleural-pressure-related complications during thoracentesis [13,18,43]. Furthermore, study data on diagnostic applicability have been sparse and not up to current reporting standards. To our knowledge, these data have only been presented in the studies by Lan et al. [7] (*n* = 65) and Halford et al. [10] (*n* = 113).

Our finding of 14/49 (28%) patients with suspected or known MPE being diagnosed with NEL is consistent with the results of other studies [5,35]. It has been suggested that many suspected iatrogenic pneumothoraxes seen after thoracentesis are, in fact, “pneumothorax ex vacuo” in the context of NEL, the process of which is believed to be caused by the drop in pleural pressure, resulting in small, transient tears or pleural-pressure-dependant fistulas in the visceral pleura with subsequent accumulation of air into the pleural space to allow for pressure equalization [1,44]. Being able to identify these patients up front will also allow for procedure avoidance, as attempts to conduct prolonged pneumothorax and suction treatment in an effort to fully expand the lung may be avoided. By utilizing the cut-off values suggested in Table A3 for lung and diaphragm movement, patients who are candidates for indwelling pleural catheters can be identified and planned more easily, and care can be taken for the patients at risk of thoracentesis-related symptoms.

In our study, measurements were performed at the “site of planned thoracentesis”. Although TUS appearances which could have more readily identified areas of suspected NEL may, therefore, have been missed, we believe that this approach is more typical of usual clinical practice and, thus, allows for greater external validity.

The ultrasound measurements utilized by M-mode and B-mode were simple to perform, ensuring reproducible and externally valid results. Measurements and cut-points were presented down to 1/100 cm, which intuitively presented a degree of uncertainty of the reproducibility for clinicians, and we propose that a validation cohort present cut-points at 1/10 cm.

SWE did not result in superior diagnostic ability of NEL in our study. Obtaining usable ROI was reasonable easy. We observed that there was a high degree of measurement unreliability within the same ROI, as exemplified in Figure 1. Even though care was taken to conform to attaining optimal imaging, the patient population was frail, and, for example, breath holding proved difficult.

There are a number of limitations to our study, including having a single centre and a relatively small number of patients included. The study was not designed with a power calculation to explore the number of participants needed. This was due to the unknown nature of the diagnostic accuracy of the tests performed and the uncertainty as to the prevalence of NEL in the study population. The lack of operator reliability measures also limits the validity. However, with the COVID-19 restrictions at the time of the study, the limited personnel availability, and a frail patient group, it was decided to limit the number of health care professionals to one physician and one nurse in the procedure room per exam. Studies on more advanced ultrasound measures have shown high test–retest and interrater reliability in comparable studies [20,22], with similar small measurement sizes in mm being distinguishable. Close collaboration between KF and JKP throughout the study aimed to ensure that procedures were performed as uniformly as possible. SWE also comes with another limitation: the increased price of purchasing and limited availability of some ultrasound machines, especially lower- and mid-grade machines.

## 5. Conclusions

Our study is the first to investigate and show that thoracic ultrasound—in the hands of chest physicians—of respiratory lung movement is useful in the diagnosis of NEL. M-mode is a basic ultrasound feature; it showed promising diagnostic ability in ruling out NEL and was superior to both B-mode and the more complex and resource-demanding shear wave elastography.

Our study suggests that M-mode should be a basic thoracic ultrasound skill in the hands of respiratory physicians involved in the management of pleural diseases.

## Figures and Tables

**Figure 1 diagnostics-14-00204-f001:**
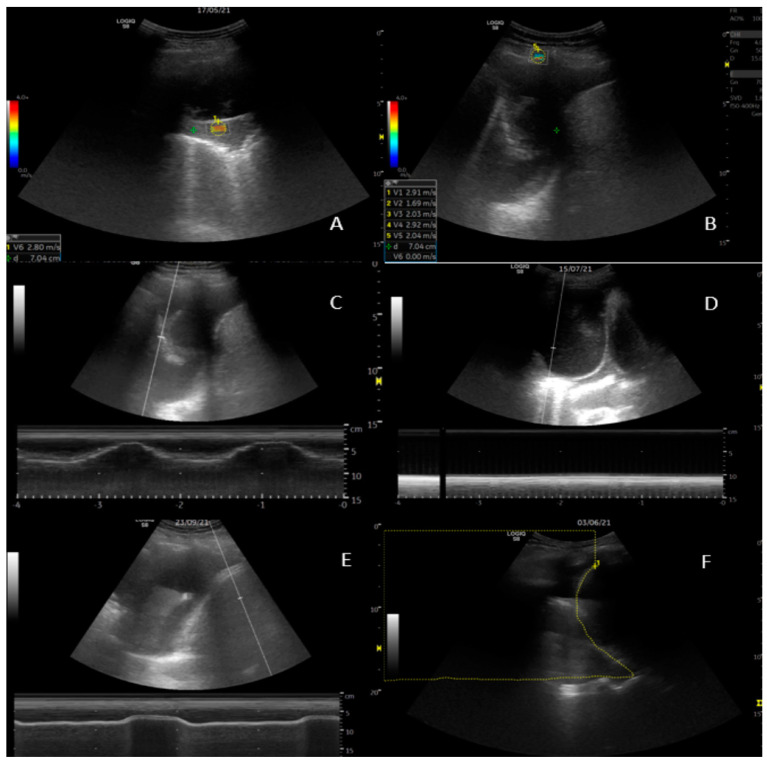
Ultrasound examinations. (**A**) 2D shear wave elastography (SWE) of visceral pleura. (**B**) SWE of parietal pleura with 5 readings recorded in the lower left corner. (**C**) M-mode measurement of lung movement. The respiratory pattern is seen in the curves on the bottom of the screen. (**D**) M-mode measurement of lung movement. Here, absence of lung movement is illustrated by the flat line below the 2-D image. (**E**) M-mode measurement of diaphragm movement. (**F**) Area-method measurement of area above the diaphragm during inspiration.

**Figure 2 diagnostics-14-00204-f002:**
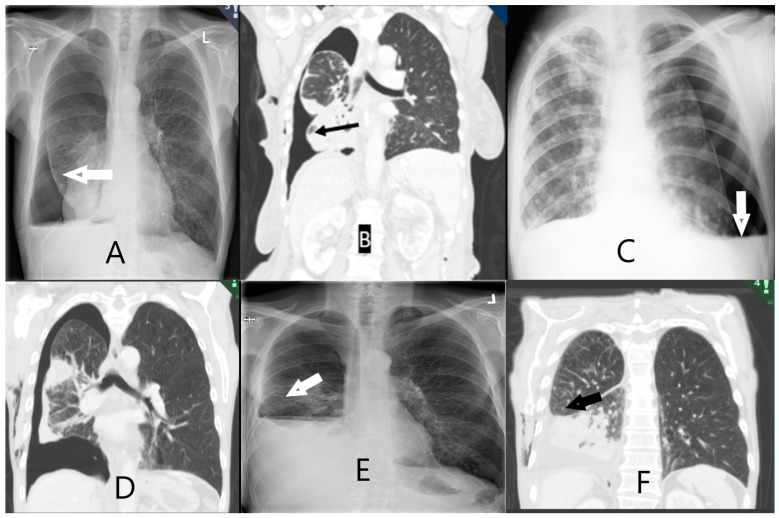
Radiological images of non-expandable lung (NEL). CXR (image (**A**,**C**,**E**)) and CT-scan (image (**B**,**D**,**F**)). (**A**) NEL. Lower lobe with thickened visceral lining. (**B**) NEL. Obstructive tumour of the lower lobe. (**C**) NEL. Hydropneumothorax following thoracentesis. (**D**) NEL with thickened pleura. (**E**) NEL. Small hydropneumothorax following thoracentesis. (**F**) NEL. Pleural effusion following thoracentesis. Fluid has replaced initial air in the pleural space seen on CXR after thoracentesis.

**Figure 3 diagnostics-14-00204-f003:**
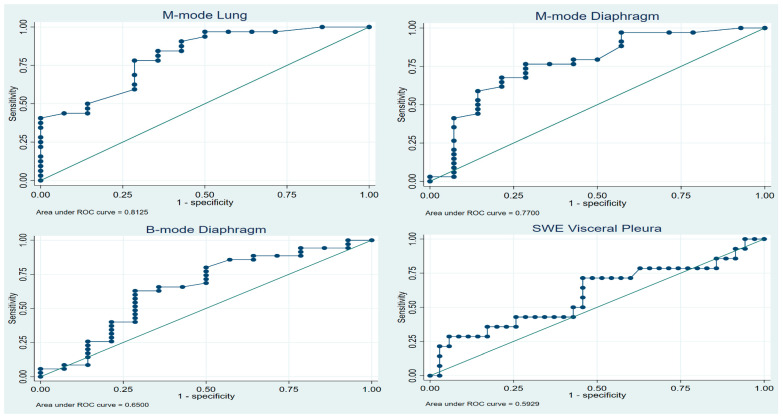
Best-performing ROC curves. (SWE *Shear Wave Elastography*).

**Table 1 diagnostics-14-00204-t001:** Demographics.

Variable	Expandable Lung	Non-Expandable Lung	*p*-Value
	*n* = 35	*n* = 14	
Age	71 (63–76)	76 (68–80)	0.071 #
Sex (female)	18 (51%)	6 (43%)	0.59 ¤
Smoking status			0.40 ¤
Current/Former	8 (23%)/19 (54%)	3 (21%)/10 (71%)	
Never	8 (23%)	1 (7%)	
Accumulated smoking pack years	34 (15–43)	28 (18–40)	0.81 #
Known malignancy	31 (89%)	13 (93%)	0.65 ¤
Pleural fluid volume drained (mL)	1290 (604)	888 (620)	0.049 ∩

Data are presented as mean (SD) or median (IQR) for continuous measures, and *n* (%) for categorical measures. #: Wilcoxon rank-sum test. ¤: Chi-square test. ∩: Two-sample *t* test.

**Table 2 diagnostics-14-00204-t002:** Ultrasound test measurements with ROC-curve characteristics.

Test Modality	Median (IQR)	AUC	95% CI
**M-mode**			
Lung movement (cm)	0.94 (0.44–1.11)	0.81	0.68–0.95
Diaphragm movement (cm)	1.21 (0.56–2.01)	0.77	0.61–0.93
**B-mode**			
Diaphragm movement (cm)	0.86 (0.41–1.27)	0.65	0.46–0.84
Area method—delta (cm^2^)	11.50 (6.69–15.95)	0.60	0.40–0.79
Lung sliding *	24 (50%)	0.74	0.61–0.87
**Shear Wave Elastography**			
Visceral pleura (m/s)	1.64 (1.47–2.08)	0.59	0.40–0.79
Pleural effusion (m/s)	1.64 (1.34–2.06)	0.53	0.34–0.71
Parietal pleura (m/s)	2.54 (2.03–3.09)	0.57	0.39–0.75

* *n* (%) for categorical measures.

## Data Availability

All data from the study are presented in this manuscript. For questions concerning data, please contact the corresponding author.

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
