# Peer review of "Ultrasound in the Diagnosis of Non-Expandable Lung: A Prospective Observational Study of M-Mode, B-Mode, and 2D-Shear Wave Elastography"

_diagnostics, 2024, doi:10.3390/diagnostics14020204_

Round 1

Reviewer 1 Report

Comments and Suggestions for Authors

Dear Editor and Authors,

It was my pleasure to be asked to review this manuscript titled “Ultrasound in the Diagnosis of Non-Expandable Lung: An Observational Study of M-mode, 2 B-mode and 2D-Shear Wave Elastography” by Dr. Petersen and his colleagues in Denmark.

In this single institution, prospective observational pilot study the authors investigate a very interesting hypothesis; that thoracic ultrasound can be used to predict a lung that may not expand following thoracocentesis and drainage of a pleural effusion. As a thoracic surgeon involved in pleural disease management the concept is intriguing. Although this hypothesis has been floating around the authors have performed and exploratory study on it! More specifically they have utilized Ultrasound 2D Shear Wave Elastography (SWE) as their focused measurement to establish NEL as well as M-mode and B-mode. They concluded that “M-mode showed promising diagnostic ability in ruling outNEL, and was superior to both B-mode and the more complex and resource-demanding shear wave elastography”.

The study’s methodology appears robust with clear inclusion and exclusion criteria, well defined procedure and assessment parameters. In addition all pertinent ethical requirements have been observed. Thus the study seems well performed. The manuscript is clear and concise in easy to understand language and it is also well illustrated with tables and figures.  

I only have some minor comments to offer, thank you.

Comments:

1.       Line 159 needs some clarification because it reads like the radiologists assessed images taken at four months and not those obtained right after thoracocentesis! If images to establish NEL were taken at 4 months as the authors very well know the space would have filled in again! I believe the authors mean to say that the initial images/CXR post hemithorax drainage were eventually assessed at 4 months!! Please clarify this in your text and tables!!!

2.       The number of patients studied is actually quite small at 14!! This should make this a pilot study at best! Did the authors perform a sample size calculation prior to enrollment and analysis to ascertain if they have statistically meaningful results?

3.       Why did the authors not present the type of malignancy the patients have in their demographics table? Do they not have this information? The type of malignancy (mesothelioma, primary lung cancer, metastatic cancers – ovarian, GI, renal ect) plays a role in terms of how they behave and develop!  

Comments on the Quality of English Language

Needs some minor language editing but nothing major.

Author Response

Dear reviewer, we thank you for your very sound comments.

  1. We agree that the line can be read as you say, and yes, the images assessed were taken shortly after thoracentesis were reviewed at the follow up date. We will adjust the text to clarify.

  1. A number of factors were assessed prior to the statistical plan. As these diagnostic tools were not applied to diagnose NEL beforehand we did not know what to expect from sensitivity and specificity - we initially planned for both a development and validation cohort, but COVID-19 and a timely restricted PhD research period forced us to stop inclusion earlier. A power calculation would require assumptions on the diagnostic accuracy of the tests (which though were not available). Moreover, we were unsure on the prevalence of patients with NEL in our population.

  1. The total number of patients who had a diagnosis of malignancy at follow up was 13/14 patients with 6 patients with lung cancer, 4 with mesothelioma, 2 breast cancer and one with prostate cancer. We discussed if we should include the diagnosis but were afraid that patient GDPR integrity could be violated. We agree that the behaviour and symptoms of patients can vary secondary to the underlying malignancy. We have included the most common malignant causes and grouped the two smallest groups in the revised document. 

Reviewer 2 Report

Comments and Suggestions for Authors

The authors Petersen and company describe a prospective observation study on the use of ultrasound to determine if a patient has NEL. They make good arguments for the importance of this topic and support plausible physiological premise for their their study. This is a well written manuscript. I think that the authors have sound methodology and well thought out statistical plan. They provide clear cut results with a high quality discussion on current literature and the use of TUS and the importance and relevance of their study. 

Furthermore, they describe the uniqueness, although I am not entirely convinced that at the hands of a cardiology trained versus pulmonary trained ultrasonographer when it comes to this evaluation is that different. I would assume that M-mode is not "terribly" advanced ultrasound technique relegated to cardiologist. Regardless of my personal opinions, I believe this current study adds to the body of literature for both the continual recognition of NEL and the cause of pneumothorax and furthermore promotes the utility and use of TUS beyond just evaluating "where to stick the needle." 

I have only some minor questions and comments, and should not really tie up the publication process.

1. Why was benign etiologies excluded?

2. Would the authors be able to tell us which patients had their follow-up NEL determined by CXR and which were determined by CT Scan? It has been some of my own practice experience when doing pleural manometry that when finding a pleural elastance at the proposed cut-off, that the final chest radiographic, the front CXR is usually concluded to be "normal," but with a follow-up Chest CT it does reveal a small, although in consequential pneumothorax. I wonder if the author's have had similar experiences in the past.

3. Lines 306-307 "the process of which is believed to be caused by the drop in 306 pleural pressure, resulting in small, transient tears in the visceral pleura and subsequent 307 accumulation of air into the pleural space to allow pressure equalisation(1)." It should be noted, this idea of a air tight visceral pleura is being challenged, and maybe it's not "tears" in the visceral pleura, but rather the development of parenchymal pleural fistulas. I do find it interesting how a change in this theory is voiced by the same authors of the reference cited by the authors. Here is the https://doi.org/10.1016/j.chest.2023.04.049.

Author Response

Dear Reviewer, thank you for taking the time to access our study. We appreciate your well written comments and are delighted that you find the study acceptable to go forth in the publication process.

We by no means meant to insult cardiologist – the diagnostic tools we used are in fact more simple than the ultrasound methods used by the Salamonsen study  (except maybe shear wave elastography which proved somewhat volatile and susceptible to outer factors such as breathing and heart pulse).

  1. Benign etiologies were excluded to gain a more homogenous population. The literature describes many causes leading to NEL, some reversible. In some studies, upward of 30% in a population with malignant pleural effusion have been found to have NEL. As our study population was expected to be limited, we wanted to achieve a reasonable powered study, without being able to perform a power calculation because of the unknown size of the sensitivity and specificity of the applied diagnostic tests applied as well as the disease prevalence.
  2. We did not collect data specifically on what the first applied radiological examination was. Most patients had a CXR following thoracentesis. In preparing the CXR and CT-scans for the assessors, less than a handful of patients went directly to CT-scans following thoracentesis as these were pre-planned in the patients’ cancer evaluation (diagnostic or treatment evaluation). A number of patients had CT-scans performed in the follow up period which the two independent assessors also had access to at follow up.
  3. Thank you for sharing this knowledge with us. This theory could very well explain both the sudden onset and stop of progression of air leaked into the pleural space – either through a tear or through fistulas – the originally proposed pleural visceral tears might as well be pressure dependent fistulas – hopefully further research will bring us closer to an even better understanding of non-expandable lung - we have incorporated this new knowledge in our discussion.